# Selection of Enzymatic Treatments for Upcycling Lentil Hulls into Ingredients Rich in Oligosaccharides and Free Phenolics

**DOI:** 10.3390/molecules27238458

**Published:** 2022-12-02

**Authors:** Sara Bautista-Expósito, Albert Vandenberg, Montserrat Dueñas, Elena Peñas, Juana Frias, Cristina Martínez-Villaluenga

**Affiliations:** 1Department of Technological Processes and Biotechnology, Institute of Food Science Technology and Nutrition (ICTAN-CSIC), José Antonio Novais 10, 28040 Madrid, Spain; 2Department of Plant Sciences, University of Saskatchewan, Saskatoon, SK S7N 5A2, Canada; 3Research Group in Polyphenols, Unidad de Nutrición y Bromatología, Facultad de Farmacia, University of Salamanca, Miguel de Unamuno Campus, 37007 Salamanca, Spain

**Keywords:** lentil hull, enzymatic hydrolysis, microwave-assisted extraction, phenolic compounds, oligosaccharides, antioxidant activity

## Abstract

In this study, the comprehensive chemical characterization of red lentil hulls obtained from the industrial production of football and split lentils was described. The lentil hulls were rich in dietary fiber (78.43 g/100 g dry weight with an insoluble to soluble fiber ratio of 4:1) and polyphenols (49.3 mg GAE/g dry weight, of which 55% was bound phenolics), which revealed the suitability of this lentil by-product as a source of bioactive compounds with recognized antioxidant and prebiotic properties. The release of oligosaccharides and phenolic compounds was accomplished by enzymatic hydrolysis, microwave treatment and a combination of both technologies. The key role played by the selection of a suitable enzymatic preparation was highlighted to maximize the yield of bioactive compounds and the functional properties of the lentil hull hydrolysates. Out of seven commercial preparations, the one with the most potential for use in a commercial context was Pectinex^®^ Ultra Tropical, which produced the highest yields of oligosaccharides (14 g/100 g lentil hull weight) and free phenolics (45.5 mg GAE/100 g lentil hull weight) and delivered a four-fold increase in terms of the original antioxidant activity. Finally, this enzyme was selected to analyze the effect of a microwave-assisted extraction pretreatment on the yield of enzymatic hydrolysis and the content of free phenolic compounds and oligosaccharides. The integrated microwave and enzymatic hydrolysis method, although it increased the solubilization yield of the lentil hulls (from 25% to 34%), it slightly decreased the content of oligosaccharides and proanthocyanidins and reduced the antioxidant activity. Therefore, the enzymatic hydrolysis treatment alone was more suitable for producing a lentil hull hydrolysate enriched in potential prebiotics and antioxidant compounds.

## 1. Introduction

Lentil, a pulse crop ranking fourth in terms of global grain legume production, is a dietary source of protein, starch, fiber and micronutrients that are important to human nutrition [1]. The growing demand for plant-based proteins, combined with the existing knowledge of the potential health effects of increased pulse dietary intake, have raised the production of lentil products [1]. Dehulling is a primary process for producing dehulled lentil seeds or split lentils, lentil flour and fractionated protein and fiber ingredients [2]. By-products generated from dehulling are mainly hulls (8–16% of dry seed weight), embryonic axes (1–3% of dry seed weight) and broken cotyledons [2]. As a consequence, the dehulling industry generates a large amount of lower-value by-product, representing 20–28% of the total lentil amount processed [2]. The primary markets for lentil hulls are of low value and have very limited use in human nutrition. Therefore, this by-product not only represents a low-value disposal problem for millers but also wastes a potential source of novel, nutritious and health-promoting food ingredients.

Lentil hulls are a promising source of nutrients that generally contain 60–90% dietary fiber, 2–8% protein, 3% ash and 1–3% lipids [2]. Lentil dietary fiber consist of cellulose, pectin, xylans and mannans, which can be considered as sources of prebiotics [3]. Moreover, lentil hulls have large amounts of extractable phenolics (procyanidins being the major group followed by phenolic acids and flavonols) and considerable levels of conjugated and bound phenolic acid derivatives linked to cell wall components (proteins, cellulose, hemicellulose and pectin) [4]. Besides the well-documented physiological benefits of dietary fiber [5], the phenolic compounds of lentil hulls provide potential for various physiological benefits such as those related to antioxidant and anti-inflammatory activities [6,7,8]. These findings represent a good foundation for further investigation into the value-added use of these by-products, particularly in functional foods and nutraceutical products that improve health. Some challenges in the development of functional ingredients from lentil hulls include the compact inner insoluble food matrix, which may be a physical barrier to the release and absorption of phytochemicals and may contribute to a loss of efficacy in maintaining health and reducing the risk of disease [9]. It often seems to be the case that although polysaccharides are fermented by the colonic microbiota, the selectivity for health-promoting bacterial groups is increased by partial hydrolysis [10]. As the bioavailability of bioactive compounds plays an important role in the health benefits of lentil hulls, the health-outcome-oriented hydrolysis of the food matrix using physical, chemical or enzymatic methods are promising strategies for the better exploitation of this byproduct as a functional food ingredient.

Earlier research has demonstrated that the use of commercial enzymes increases the amounts of free phenolics in bran/hull byproducts obtained from certain cereals as well as causing a concurrent increase in their radical scavenging and anti-inflammatory activities [11,12]. The synergy between different feruloyl esterases, pectinases, cellulases, hemicellulases and proteases present in commercial enzymatic preparations is crucial in breaking the bonds among cell-wall polysaccharides, proteins and polyphenols [13]. Differences in the enzymatic profile of commercial enzymatic preparations make the screening and selection of enzymes a key step in the development of functional ingredients from agro-industrial by-products. The integration of physical processing methods has resulted an adequate approach towards a higher solubilization efficiency of bioactive compounds [14]. In recent years, microwave-assisted extraction (MAE) has received considerable attention because it is a green (low energy consumption, non-thermal technology) and appealing technological alternative to conventional chemical extraction processes. The high temperature and pressure involved in the process facilitate the destruction of material surfaces, which in turn results in an increased area and increased amounts of compounds that are released into the solvent [15]. Microwave treatment has been widely studied to enhance the extractability of various components such as protein, oil and bioactive components from agro-industrial byproducts, but there have been no studies that have dealt with the use of MAE for the extraction of phenolics and oligosaccharides from legume seed coats.

In this study, we hypothesized that lentil hulls could be valorized through the extraction of oligosaccharides and phenolic compounds with a high added value due to their potential biological properties. We conducted a comprehensive chemical characterization of red lentil hulls resulting from the industrial processing of red football and split lentils. Likewise, the successful extraction of oligosaccharides and phenolic compounds from the lentil hull by-products through microwave treatment, enzymatic treatment and a combination of the two was accomplished, and their potential as functional food ingredients was accordingly discussed. Seven different commercially available food-grade enzyme preparations, designed to break down plant cell walls (Novozymes A/S), were screened for their ability to increase the extraction yield of bioactive compounds and antioxidant activity.

## 2. Materials and Methods

### 2.1. Materials

Hulls from an industrial de-hulling process that produces red football and split lentils were kindly provided by Prairie Pulse Inc. (Vanscoy, Saskatchewan, SK, Canada) in May 2019. Hulls were ground into a fine powder with a mixer mill (MM 400, Retsch, Haan, Germany) for approximately 3 min at maximum speed and stored in sealed plastic bags at 4 °C. Ultraflo XL, Ultraflo Max, Ultimase BWL 40, Viscozyme L, Celluclast 1.5 L, Pectinex^®^ Ultra Tropical and Shearzyme Plus 2X were obtained from Novozymes (Bagsvaerd, Denmark). Fast Blue BB (FBBB) [4-benzoylamino-2,5-dimethoxybenzenediazonium] chloride hemi-(zinc chloride), 2,2′-azinobis 3-ethylbenzothiazoline-6-sulfonic acid (ABTS), 2,2′-diazobis-(2-aminodinopropane)-di-hydrochloride (AAPH) and fluorescein were purchased from Sigma-Aldrich Co. (St. Louis, MO, USA). Standards such as (+)-catechin, *trans-p*-coumaric acid, quercetin 3-*O*-rutinoside, quercetin 3-*O*-glucoside and kaempferol 3-*O*-rutinoside were provided by Extrasynthese (Lyon, Genay Cedex, France). Standards of 6-hydroxy-2,5,7,8-tetramethyl-2-carboxylic acid (Trolox), D-glucose, glycerol and gallic acid were acquired from Sigma-Aldrich Co. (St. Louis, MO, USA).

### 2.2. Enzymatic Treatments

Enzymatic extractions were performed using seven commercial glucanases: Ultraflo XL, Ultraflo Max, Ultimase BWL 40, Viscozyme L, Celluclast 1.5L, Pectinex Ultra Tropical and Shearzyme Plus 2X (Novozymes, Bagsvaerd, Denmark). Enzymatic treatments (100 mL) were performed in water at a solid-to-solvent ratio of 1:20 (*w*:*v*) in accordance with previous studies [11]. Reaction mixtures consisted of a 1% enzyme-to-lentil hull ratio (*w*:*w*) according to the manufacturer’s recommendations and were processed at 40 °C for 3 h in a Thermomixer C shaker (Eppendorf Ibérica, Madrid, Spain) at 2000 rpm. The reaction with Pectinex^®^ Ultra Tropical was monitored at selected times (0, 0.5, 1, 1.5, 2, 2.5, 3, 3.5 and 4 h) by taking 2 mL aliquots. Enzymes were deactivated at 95 °C in a water bath for 5 min.

### 2.3. Comparison of Microwave, Enzymatic and Sequential Microwave–Enzymatic Treatments of Lentil Hull

Microwave extraction was performed using a lentil-hull-to-solvent ratio of 1:20 (*w*:*v*) in a Pyrex bottle, to which 20 mL of bi-distilled water was added. Microwaves were applied at 700 W and 85 °C for 1 min using a JMO011138 microwave (Jocel, Argemil, Portugal) equipped with a rotating plate. The tested power settings were selected according to previous research designed to extract bioactive compounds from fruit peels and pomaces [16,17]. The extracted samples were cooled in an ice water bath for 10 min. For the sequential microwave–enzymatic extractions, the microwave-treated lentil hulls were mixed (step 1) with Pectinex^®^ Ultra Tropical (1% enzyme-to-lentil-hull weight ratio) (step 2). The reaction mixtures with a lentil-hull-to-solvent ratio of 1:20 (*w*/*v*) were incubated for 3 h at 40 °C in a thermomixer C (Eppendorf Ibérica, Madrid, Spain). Enzymes were deactivated by boiling in a water bath for 5 min.

For the three lentil hull treatments, samples were centrifuged for 10 min at 10,000× *g* in a microcentrifuge (Eppendorf AG, Hamburg, Germany), and the weight of the supernatant was recorded and subsequently freeze-dried to gravimetrically determine the dried water-soluble fraction weight.

### 2.4. Determination of Total Dietary Fiber (TDF), Insoluble Dietary Fiber (IDF) and Soluble Dietary Fiber (SDF) Fractions

The total (TDF), insoluble (IDF) and high-molecular-weight soluble dietary fiber (HMW-SDF) content in the lentil hulls was determined by enzymatic and gravimetric methods using a Rapid Integrated Total Dietary Fiber Assay Kit (K-RINTDF, Megazyme, Wicklow, Ireland). TDF, IDF and HMW-SDF were expressed as g/100 g dry lentil hull weight (DW).

Low-molecular-weight soluble dietary fiber (LMW-SDF), which represents non-digestible oligosaccharides of a degree of polymerization ≥ 3, was determined by size exclusion chromatography following the K-RINTDF protocol (https://www.megazyme.com/documents/Assay_Protocol/K-RINTDF_DATA.pdf, accessed on 10 January 2022). Dried lentil hulls and hydrolysates were dissolved in 600 µL of double-distilled water, and glycerol was added as an internal standard at a final concentration of 10 mg/mL. A high-resolution liquid chromatograph (HPLC) Alliance Separation Module 2695 (Waters, Milford, MA, USA) equipped with a 2414 refractive index detector (Waters, Milford, MA, USA) maintained at 50 °C was used. The sample injection volume was 50 μL. Separation was performed in the isocratic mode using microfiltered distilled water as the solvent, two TSKgel^®^ G2500 PWXL columns (7.8 mm id × 30; Tosoh Co., Tokyo, Japan) connected in series at a flow rate of 0.5 mL/min and at a temperature of 80 °C were used, and a run time of 60 min was used to ensure the column was cleaned out. The LMW-SDF content was expressed as g/100 g DW using the equation (1):Oligosaccharides (g/100g) = [(Rf × Wt_IS_ × PA_LMW-SDF_/PA_IS_) × (100/M)]/1000 (1)
where Rf is the D-glucose response factor (ratio of peak area of D-glucose/peak area of glycerol to the ratio of the mass of D-glucose/mass of glycerol); Wt_IS_ is the amount of internal standard contained in 1 mL of glycerol internal standard solution pipetted into the sample before filtration in mg; PA_LMW-SDF_ is the peak area of the LMW-SDF fraction; PA_IS_ is the peak area of the glycerol internal standard; and M is the test portion mass in grams of the sample analyzed by HPLC.

### 2.5. Determination of Total Protein and Starch

The total protein content of lentil hulls (0.5 g) was determined by the Dumas combustion method using a Trumac nitrogen analyzer (Leco Corporation, St Joseph, MI, USA). A conversion factor of 6.25 was used to convert the nitrogen values to protein content. The results were expressed as g/100 g DW.

Total starch was determined by an enzymatic–colorimetric method using a K-TSTA-100A Total Starch Assay Kit (Megazyme, Wicklow, Ireland). A Synergy HT microplate reader (BioTek Instruments, Winooski, VT, USA) was used to read the absorbance at 510 nm. The results were expressed as g/100 g DW.

### 2.6. Determination of Anti-Nutrients

Trypsin inhibitory activity (TIA) was determined as previously reported [18]. Briefly, samples (100 mg) were extracted in 5 mL of 0.01 M NaOH (pH 8.4–10.0) and shaken for 3 h at 20 °C in a Thermomixer C (Eppendorf, Thermo Fisher Scientific, Waltham, MA, USA) at 2000 rpm. Extract volumes were adjusted to 10 mL with distilled water, shaken for 1 min at 2000 rpm and left standing for 15 min. Aliquots of 1 mL were withdrawn and diluted to produce a 40–60% inhibition of the trypsin activity. TIA was expressed as trypsin inhibitory units (TIU)/mg DW.

Phytic acid content was determined by an enzymatic–colorimetric method using a K-PHYT Phytic Acid (Phytate)/Total Phosphorus Assay kit (Megazyme, Wicklow, Ireland). Absorbance was read at 655 nm using a Synergy HT microplate reader (BioTek Instruments, Winooski, VT, USA). Phytic acid content was expressed as g/100 g DW.

Condensed tannins were determined as described previously [18]. Briefly, 200 mg of the sample was hydrolyzed with 10 mL of hydrochloric acid (HCl)/n-butanol (5:95, *v*:*v*) containing 0.7 g/L of ferric (III) chloride (FeCl_3_) at 100 °C for 1 h. Samples were centrifuged (14,000× *g* for 10 min), and the supernatants were washed two times with 10 mL of n-butanol:HCl:FeCl_3_. After adjusting the final volume to 25 mL, absorbance was measured at 550 nm in a Synergy HT microplate reader (BioTek Instruments, Winooski, VT, USA). The condensed tannin content was expressed as mg catechin equivalents (CAE)/g DW using a catechin calibration curve.

### 2.7. Extraction of Free and Insoluble Phenolic Compounds

Extraction of the free phenolic compounds of the lentil hulls was performed according to a previously described method [19] with some modifications. Lentil hull dry powder (1 g) was extracted with 20 mL of 80% ethanol, vortexed for 30 s by an IKA Vortex 3 (IKA^®^-Werke GmbH & Co, Staufen, Germany) and shaken for 10 min at 4 °C and 1500 rpm in a Thermomixer C (Eppendorf, Hamburg, Germany). The extract was centrifuged (Centrifuge 5424 R; Eppendorf AG, Hamburg, Germany) at 2500× *g* and 4 °C for 10 min, and the supernatant was transferred into a new labeled tube. The residue was submitted to a second cycle of extraction in the same conditions. The combined supernatant containing free phenolic compounds was evaporated by a rotary vacuum evaporator at 40 °C (Rotavapor^®^ R-300, Büchi Labortechnik AG, Flawil, Switzerland) and re-suspended in 2 mL of absolute methanol.

The dry residue remaining (1 g) after the removal of free phenolics was subjected to alkaline and acid hydrolysis. The residue was re-suspended in 12 mL of 2 M NaOH, vortexed for 30 s and stirred overnight at room temperature. The hydrolysate was acidified with 6 N HCl to pH 2, and the released phenolics were extracted with 7 mL of ethyl acetate three times and centrifuged at 10,000 *×* g and 4 °C for 10 min. The remaining aqueous layer was subsequently hydrolysed with 2.5 mL of 6 N HCl and incubated at 85 °C for 30 min. After cooling down on ice for 5 min, hydrolysates were repartitioned with ethyl acetate three times. The organic layers of alkaline and acid hydrolysates were evaporated by a rotary vacuum evaporator at 40 °C (Rotavapor^®^ R-300, Büchi Labortechnik AG, Flawil, Switzerland), reconstituted in 5 mL of 70% methanol and filtered using 0.22 µm syringe filters prior to analysis.

### 2.8. Determination of Total Free and Insoluble Phenolic Content

The free and insoluble phenolic content was determined by the FBBB reaction according to [20]. Briefly, 1mL of lentil extract or standard was mixed with 100 μL of freshly prepared FBBB reagent (0.1% in distilled water) and vortexed for 1 min. Immediately, the extract or standard solutions were shaken after adding 100 μL of 5% NaOH and allowed to react for 120 min at room temperature. Finally, 200 μL of the reaction mixture was placed in a 96-well plate, and the absorbance was measured at 420 nm using a Synergy HT (BioTek Instruments, Winooski, VT, USA) microplate reader. Quantification of the polyphenol content was performed using a gallic acid calibration curve (0–225 μg/mL, R^2^ > 0.99). All analyses were performed in duplicate. The data were expressed as mg of gallic acid equivalents (GAE)/g DW.

### 2.9. Analysis of Phenolic Profile by HPLC-DAD-ESI-MS^2^

The identification and quantification of free phenolic compounds was performed according to a previously described method [21]. First, purification of the phenolic extracts was carried out by solid-phase extraction using C18 Sep-Pak cartridges (Waters, Milford, MA, USA), which were previously activated with methanol (2 mL), followed by distilled water (3 mL). Purified sample extracts were injected onto a Hewlett–Packard 1100-diode array detector (DAD) liquid chromatograph (Agilent Technologies, Palo Alto, CA, USA) including a quaternary pump. The mobile phases utilized were 0.1% formic acid in water (solvent A) and 100% acetonitrile (solvent B). The elution gradient employed was 15% B for 5 min, 15–20% B for 5 min, 20–25% B for 10 min, 25–35% B for 10 min, 35–50% B for 10 min and column re-equilibration. The chromatographic separation of the phenolic compounds was conducted at a flow rate of 0.5 mL/min at 35 °C in a Spherisorb S3 ODS-2 C8 column (Waters, Milford, MA, USA) (3 μm, 150 mm × 4.6 mm i.d.). Based on the different maximum absorbance wavelengths among the phytochemicals, the preferred wavelengths for DAD detection were 280 nm (hydroxybenzoic and hydroxycinnamic acids) and 370 nm (flavonols). The mass spectrometer (MS) was coupled to the HPLC system through the DAD cell output, and the detection was conducted in an API-3200 Qtrap (Applied Biosystems, Darmstadt, Germany) equipped with an ESI source, a triple quadrupole-ion trap mass analyzer and the Analyst 5.1 software. The phenolic compounds were identified using their retention times, UV and mass spectra, fragmentation patterns and comparison to authentic standards when available. For quantitative analysis, gallic acid and *trans-p*-coumaric derivatives were measured using the calibration curves of the respective free acid. Quercetin derivative was quantified using the respective quercetin-3-*O*-glucoside curve, kaempferol derivatives were quantified using kaempferol 3-*O*-rutinoside and (+)-catechin and proanthocyanidins were quantified using the (+)-catechin curve and (−)-epicatechin was quantified using the (−)-epicatechin curve. The concentrations of each phenolic compound were expressed as µg/g DW.

### 2.10. Oxygen Radical Absorption Capacity (ORAC) Assay

An ORAC assay of the lentil hull extracts and hydrolysates was determined following an earlier reported procedure [18]. Briefly, 180 μL of 70 nM fluorescein was mixed with 90 μL of 12 mM AAPH and 30 μL of extract or standard. The reaction mixtures were placed in a black 96-well plate (Fisher Scientific, Waltham, MA, USA), and the fluorescence was measured in a Synergy HT microplate reader (BioTek Instruments, Winooski, VT, USA) every minute at excitation and emission wavelengths of 485 and 520 nm, respectively. An external calibration curve using Trolox as the standard in a linear concentration range from 0 to 160 μM was prepared from a freshly made 1 mM stock solution. The results were expressed as µmol Trolox equivalents (TE)/g DW.

### 2.11. ABTS (2,2′-Azinobis 3-ethylbenzothiazoline-6-sulfonic acid) Radical Scavenging Assay

The ABTS radical scavenging activity of the lentil hull extracts and hydrolysates was measured following a previously reported method [22]. Briefly, an ABTS radical (ABTS^•+^) solution was prepared by mixing 7 mM ABTS with 2.45 mM K_2_O_8_S_2_ at a 1:1 (*v*/*v*) ratio. The mixture was reacted for 16 h (room temperature, dark conditions). Then, the absorbance at 734 nm of the ABTS^+^ working solution was adjusted to 0.70 ± 0.02 by diluting with phosphate buffer (75 mM, pH 7.4). A volume of 20 μL of the extract or standard was mixed with 200 μL of the ABTS^+^ working solution in a 96-well microplate. The mixtures were reacted for 30 min in darkness at room temperature. The absorbance was read at 734 nm in a Synergy HT microplate reader (BioTek Instruments, Winooski, VT, USA), and a Trolox calibration curve was used in the concentration range from 0 to 800 μM. The results were expressed as µmol TE/g DW.

### 2.12. Statistical Analysis

All the replicated chemical composition analyses and the oligosaccharide, phenolic compounds and water-soluble fraction yield assays were repeated twice. The experimental data were expressed as the mean and standard deviation of six values accordingly. Pearson’s correlation was performed to elucidate the relationships among the variables. The differences between the experimental groups were compared by one-way analysis of variance (ANOVA) and Bonferroni’s post hoc test. Differences with *p* ≤ 0.05 were considered statistically significant. All statistical analyses were conducted using Statgraphics Centurion XVIII (Statgraphics Technologies, The Plains, VA, USA).

## 3. Results

### 3.1. Red Lentil Hulls Are a Source of Dietary Fiber and Polyphenols, Containing Considerable Amounts of Protein and Trypsin Inhibitors and Minor Amounts of Starch and Phytic Acid

The assessment of the nutritional composition of lentil hulls is essential for the determination of any potential use in the development of value-added products. The chemical composition of the red lentil hulls obtained from an industrial lentil-processing plant is shown in Table 1. Almost 90% of the red lentil hulls were composed of carbohydrates and proteins. Among the carbohydrates, TDF was the main compound, representing 78.4 g/100 g DW, which was a slightly higher value compared to those of earlier studies (73.34 and 71.32 g/100 g DW for red and green lentil hulls, respectively) [23]. Dietary fiber is an important food ingredient with prebiotic properties. Properly increasing dietary fiber intake can increase gastrointestinal motility, improve the abundance of the beneficial gut microbiota and promote the production of short-chain fatty acids so as to prevent gastrointestinal or related diseases [5]. The results of previous studies revealed that only 3.05%, 3.09%, 4.37% and 2.82% of the TDF in the seed coats of mung bean, fava bean, lentil and pea, respectively, was soluble [24]. This was in agreement with our results, which showed insoluble non-starch polysaccharides as the most abundant dietary fiber components in the red lentil hulls (IDF, 69.3 g/100 g DW) with minor amounts of soluble non-starch polysaccharides (HMW-SDF, 9.11 g/100 g DW, Table 1). The HMW-SDF content in the red lentil hulls reported herein was notably higher than that previously reported for red and green lentil hulls (1.5 and 2.90 g/100 g DW, respectively) [23]. Non-digestible oligosaccharides (LMW-SDF) were not detected in the red lentil hulls (Table 1) in accordance with Zhong et al. [2].

The ratio of insoluble and soluble fiber may play an important role in influencing the potential health benefits of products. The IDF:HMW-SDF ratio of the red lentil hulls was 7.6, indicating that only about 14% of the intrinsic dietary fiber in the lentil seed coat was accessible for fermentation by the gut microbiota. This was a relatively low content of microbiota-accessible carbohydrates, compared with the 75–90% value for the intrinsic dietary fiber of fruits and vegetables [5]. Thus, the food processing of lentil seed coats through changing the physical structure of dietary fiber may have great potential for improving the quantity and quality of microbiota-accessible carbohydrates.

Regarding the protein content, the mean value reached 9.12 g/100 g DW, which was very close to the reported values for red (8.64 g/100 g) and green (8.58 g/100 g) lentil hulls [23]. Legume seed coats contain structural proteins such as proteoglycans and glycoproteins in the cell wall, which are used for the aggregation and expansion of cells during growth [5]. Minor amounts of starch were present in the lentil hulls (0.13 g/100 g, Table 1), which was in accordance with the literature that points out that the starch content in lentil hulls is below 10% [24].

The phytic acid content of the red lentil hulls (0.06 g/100 g DW) (Table 1) was similar to the reported values for chickpea hulls (0.08 g/100 g DW) but was markedly lower compared to the previous reports for green lentil, red lentil, fava bean and pea hulls (0.15–0.17 g/100 g) [23]. The variation in the phytic acid content of lentil hulls reported in the literature is attributed to genetic and environmental factors [25]. As compared with dehulled red lentil seeds (1.12 g/100 g DW) [18], the lentil hulls in the current study had a lower phytic acid content. In legume seeds, the majority of the phytic acid (more than 95%) is stored as globoids, which are compartmentalized in the protein storage vacuoles of the cotyledons [25], which explains the low values observed.

Trypsin inhibitory activity was detected in the red lentil hulls, although the mean values (18.3 TIU/mg DW) were remarkably high when compared to the reported values for dehulled red lentil and the whole seeds of grey zero-tannin lentils, black lentil and fava bean (10–16 TIU/mg) [18]. Typical inhibitors of gastrointestinal proteases are protein molecules such as the Bowman–Birk and Kunitz trypsin inhibitors, which are mainly located in cotyledons; however, it is known that polyphenols also have the ability to inhibit digestive enzymes such as trypsin [26]. The higher abundance of free polyphenols in lentil seed coats and their partial extraction in aqueous solutions [27] could explain the high trypsin inhibitory activity observed for the red lentil hulls obtained in the present study (Table 1).

The total phenolic content obtained as the sum of the free and insoluble phenolic fractions of the red lentil hulls was 49.8 mg GAE/g (Table 1). This high amount of phenolic compounds suggested that this lentil milling by-product was a valuable source of phenolic compounds compared to other agri-food by-products (cranberry pomace: ~13.55–15.17 mg GAE/g DW; grape pomace: 38.7 ± 0.36 mg GAE/g DW; and blueberry pomace: ~17.76–20.82 mg GAE/g DW) [28]. The TPC of the red lentil hulls observed in the present study was in the range reported in the literature (40.8–85.37 mg GAE/g DW) for typical hulls of the red, green and black lentil varieties [4,29]. The free and bound phenolic fractions were evenly distributed in the red lentil seed coats studied, with a slightly higher proportion of the bound (55% of the total phenolic content) as compared to the free phenolic fraction (45% of total phenolic fraction). Similarly, black and green lentil hulls showed a ratio of the free-to-insoluble phenolic fraction of 1:1 and 1:1.7, respectively [29]. Condensed tannins accounted for 15.83 mg CAE/g, suggesting that 75.4% of the free phenolic content in the red lentil hulls were catechins or tannins (Table 1). This finding was in agreement with earlier studies showing that procyanidins were the main phenolic subgroup (73–79%) of the free phenolic fraction of lentil hulls [4]. Regarding the distribution of the free and insoluble fractions in the total phenolic content, a variation depending on the lentil variety was found in the literature.

### 3.2. Free Phenolic Compounds in Lentil Hulls Are Major Contributors to the Antioxidant Activity of Lentil Hulls

The antioxidant activity of the free, bound and total phenolic fractions was measured by in vitro chemical assays such as ORAC and ABTS assays (Figure 1).

The antioxidant activity, as determined by the ORAC and ABTS assays, was significantly higher in the free phenolic fraction as compared to the bound fraction of the red lentil hulls (*p* ≤ 0.05), which was in agreement with earlier investigations. For instance, Sun et al. [4] demonstrated a higher antioxidant capacity, as determined by ORAC, DPPH and FRAP assays, in the free phenolic fraction than in the conjugated and bound phenolic fractions of ADM red, Laird, CDC Greenland and Eston lentil hulls. Similar results were reported by Yeo and Shahidi [6] for CDC Greenland, CDC Invincible (green), 3493–6 (green) and CDC Maxim (red) lentil hulls. The untargeted metabolomics approach performed by [8] for correlating seed coat polyphenol profiles with antioxidant activity concluded that, regardless of the pulse crop, the antioxidant activity was largely attributed to proanthocyanidins (the main phenolic subgroup found in the soluble fraction of lentil hulls, Table 1), although flavan-3-ols were also important. These phenolic compounds form hydrogen bonds with the polar head groups of the liposome phospholipids of liposome membranes, which protect against induced oxidative damage [30].

### 3.3. Pectinex^®^ Ultra Tropical Released High Amounts of Oligosaccharides and Phenolics from Lentil Hull Food Matrix

To maximize the health benefits of the lentil hulls, enzymatic hydrolysis was explored in the present study as a processing strategy for improving the quantity and quality of microbiota-accessible carbohydrates and phenolics. Increased amounts of accessible oligosaccharides and phenolics in lentil hull hydrolysates can positively influence the regulation of intestinal immunity by directly binding to the toll-like receptors on monocytes, macrophages and intestinal cells in order to modulate cytokine production and immune cell maturation in a microbiota-independent manner [5]. Bioprocessing using enzymes has been used to modify the physical structure of cereal brans and improve the release of bound phenolic acids and soluble carbohydrates [11,20,31]. However, to the best of our knowledge, there have been no studies reporting the release of oligosaccharides and phenolics from lentil hulls by means of enzymatic hydrolysis. In the present study, seven commercial enzyme preparations were used and screened for the release of oligosaccharides and phenolic compounds and their effects on increasing the antioxidant activity of the lentil hulls (Table 2).

Regardless of the enzyme type used, the enzymatic hydrolysis allowed the lentil hulls to be enriched in oligosaccharides compared to the control sample, where these carbohydrates were not detected (Table 2). The enzymatic treatment of the lentil hulls with carbohydrate-hydrolyzing enzymes broke down the cellulose, hemicellulose and pectin in the lentil hulls into smaller poly-, oligo-, di- and monosaccharides, thereby making them water-soluble and subsequently allowing their release from the lentil hull matrix.

The highest oligosaccharide concentration was observed in the hydrolysates obtained using Pectinex^®^ Ultra Tropical (14.17 g/100 g DW). Shearzyme^®^ Plus 2X (9.79 g/100 g DW) and Viscozyme^®^ L (7.54 g/100 g DW) also produced oligosaccharides but to lesser extent than Pectinex^®^ Ultra Tropical (Table 2). The rest of the enzymes tested also produced oligosaccharides, although to a lesser extent (1.55–3.57 g/100 g). The enzymatic activities of the tested commercial enzyme preparations were highly variable, and the side activities had a strong impact on the oligosaccharide yields of the hydrolysates. Nevertheless, our results suggested that the greatest increase in the oligosaccharide content of the red lentil hull hydrolysates was determined using the enzyme preparations, which were mainly pectinases. Pectinase^®^ Ultra Tropical had pectinase as its main declared activity, and it also had, to lesser extent, cellulase, xylanase and endo-1,4-β-glucanase (Appendix A). In addition, compared with the other commercial pectinase preparations, Pectinase^®^ Ultra Tropical exhibited the highest pectin-depolymerizing activity [31]. The activity of pectinase is due to a mixture of the following exo- and endo-enzymes: (1) pectinesterases, which catalyze the release of the pectin methyl-ester groups producing polygalacturonic acid and methanol; (2) polymethylgalacturonase and polygalacturonase, which break down pectin α-1,4-glycosidic linkages; and (3) polymethylgalacturonate lyases and polygalacturonate lyases, which hydrolyze pectin α-1,4-glycosidic linkages by a *trans*-elimination mechanism [32]. The highest oligosaccharide yield obtained by the Pectinase^®^ Ultra Tropical treatment (Table 2) suggested that this enzyme treatment mainly catalyzed the hydrolysis of pectin in the red lentil hulls to release pectin oligosaccharides. Like in other legume seed coats, pectin is a major component of the soluble dietary fiber in lentil, which is characterized as a multi-branched structure composed of linear 1,3- and 1,5-arabinan and linear 1,4-glucan containing arabinose, glucose and galacturonic acid as constituent sugars [33]. The partial hydrolysis of pectin and incomplete monomerization has also been reported during the Pectinex^®^ Ultra SPL enzyme-aided cell wall disintegration of carbohydrate-rich byproducts obtained from the production of concentrate soy protein [34].

Similarly, the phenolic extraction yield showed a significant variation among the enzymatic treatments that ranged between 54% and 99% of the total bound phenolic content (Table 2). Pectinex^®^ Ultra Tropical and Viscozyme^®^ L produced the highest concentration of soluble phenolics (45.5 and 41.4 mg GAE/g, respectively), whereas Shearzyme^®^ Plus 2X was the least efficient enzymatic treatment in terms of the solubilization of bound phenolics from the red lentil hull food matrix (*p* ≤ 0.05). The higher phenolic extraction yield observed when the lentil hulls were treated with Pectinex^®^ Ultra Tropical and Viscozyme^®^ L might be explained by the enhanced cell-wall structure breakdown as a result of cell-wall component hydrolysis, particularly the glycosidic bonds/linkages between phenolic compounds and cell-wall polysaccharides [16]. Similar results were reported recently by other researchers who treated cranberry pomace (59.93 g/100 g DW of IDF and 12.74 g/100 g of SDF) with different commercial enzymes to modify the dietary fiber composition and technological properties of this by-product [28]. In this previous study, Viscozyme^®^ L and Pectinex^®^ Ultra Tropical enhanced to a greater extent the yields of oligosaccharides (7.1 and 5.9 g/100 g, respectively, vs. 1.9 g/100 g for the control) and phenolic compounds (7.8 and 7.4 mg GAE/g, respectively, vs. 7.0 mg GAE/g for the control). Moreover, it was demonstrated that water-soluble fractions showed prebiotic activity and enhanced the growth of *Lactobacillus acidophilus* DSM 20,079 and *Bifidobacterium animalis* DSM 20,105 after 24 h of fermentation.

Regardless of the enzyme used, all the lentil hull hydrolysates showed between a three- to four-fold increase in antioxidant activity as compared to the control (105.11 and 117.12 µmol TE/g for the ORAC and ABTS assays, respectively). Pearson’s correlation analysis showed a significant positive association between the soluble phenolic content and antioxidant activity of the lentil hull hydrolysates (R^2^ = 0.64 and 0.74 for the ORAC and ABTS assays, respectively, *p* ≤ 0.05). Therefore, the increased antioxidant activity in hydrolysates could be attributed to the enzymatic release of phenolic compounds from the lentil hull food matrix. Moreover, the antioxidant increase could be attributed to the release of oligosaccharides, as previously reported for pectin oligosaccharides [35]. Comparisons among the enzymatic treatments indicated that Ultraflo^®^ XL, Ultimase^®^ BWL 40, Viscozyme^®^ L and Pectinex^®^ Ultra Tropical produced lentil hull hydrolysates with a higher antioxidant activity as compared to Ultraflo^®^ Max, Celluclast^®^ and Shearzyme^®^ Plus 2X (*p* ≤ 0.05). These results were strongly supported by De Camargo et al. [36] and Gómez-García et al. [37] who studied how enzymatic hydrolysis affected the antioxidant potential of winemaking by-products, in which they found a significant increase in total phenolic compounds and antioxidant activity overall when enzymes such as Viscozyme^®^ L, and Pectinex^®^ Ultra Tropical were used.

Taking into account all the results from the screening of the seven commercial enzymes, Pectinex^®^ Ultra Tropical was selected because it exhibited the highest oligosaccharide and soluble phenolics yield, which was associated to a higher antioxidant activity. To provide insight into the rate at which the release of oligosaccharides and phenolics and the increase in antioxidant activity occured, the time course of lentil hull hydrolysis by the Pectinex^®^ Ultra Tropical treatment is shown in Figure 2.

The oligosaccharide yield gradually increased in the lentil hull hydrolysates and peaked at 3 h, where the maximum yield was 14 g/100 g lentil hull hydrolysate (Figure 2A). The kinetics of phenolic extraction was much faster (Figure 2B) compared to the behavior of the oligosaccharide extraction yields (Figure 2A). The yield of phenolic compounds increased rapidly up to two-fold in 0.5 h and reached an equilibrium from 0.5 to 2 h, after which it increased again from 2 to 3 h, where the maximum yield was observed. The antioxidant activity, as determined by the ORAC and ABTS assays, showed a similar trend to the phenolic extraction yield. As a result, in this investigation, an enzymatic hydrolysis endpoint of 3 h was selected as appropriate since a reasonable amount of phenolics was extracted, and maximum oligosaccharide yields and antioxidant activity were achieved.

### 3.4. Pectinex Treatment of Lentil Hull Increased the Bioactive Compounds and Antioxidant Activity to a Greater Extent than Microwaves or Their Combined Treatment

A recent study proved that microwaves (MW) might be considered as an efficient environmentally friendly method for the extraction of phenolic compounds and for the enhancement of the antioxidant activity of pulse extracts [15]. MW technology is advantageous not only due to its quick heating ability for extraction, reduced thermal gradients, higher yield, low usage of solvent, small size of equipment and lower energy but also because of the applicability of water as a green and polar solvent [38]. Moreover, MW technology has already been tested in a semi-industrial scale for the valorization of agro-industrial by-products [39], reducing the environmental and economic impacts and promoting the beneficial effects for human health. In this work, the potential of MW to release oligosaccharides and polyphenols from lentil hulls was tested and compared with the Pectinex^®^ Ultra Tropical (EH-P) treatment and the sequential microwave–enzymatic (MW + EH-P) treatments (Table 3).

As compared to the control, the MW treatment significantly increased the amounts of oligosaccharides (4.65 g/100 g DW) and phenolic compounds (36.28 g/100 g DW) and, consequently, the water-soluble yield (12.59 g/100 g DW) in the lentil hulls (Table 3, *p* ≤ 0.05). Similarly, Andreani and Karboune [40] reported the microwave-assisted enrichment of extracts from American cranberry pomace with oligosaccharides, with a degree of polymerization yield from 7 to 10. The increased oligosaccharide levels upon the MW treatment of the lentil hull may be explained by the changes in dietary fiber composition as reported previously for whole legume seeds [41]. In black grams, chickpeas, lentils and red and white kidney beans, MW treatment resulted in a reduction in hemicellulose and cellulose content due to their breakdown into low-molecular-weight carbohydrates (oligo-, di- and monosaccharides). Likewise, microwave cooking has been shown to increase the total soluble phenolic content of whole fava bean, lentil and pea seeds [42]. Electromagnetic waves produce dipoles and ion conduction that heat solvent molecules. Microwave heating might cause severe cell rupture due to the thermal expansion of cellular liquid, the denaturation of the cell membrane and through its impact on cell wall components, to which some bioactive compounds (i.e., phenolics) might have been bound to [43].

The higher amounts of bioactive compounds (14.17 g oligosaccharides/100 g DW, 45.51 g GAE/g DW) and the higher water-soluble fraction (24.92 g/100 g DW) and antioxidant activity (424–442 µmol TE/g DW) found in the lentil hull hydrolysates indicated that, in our study, the EH-P treatment was more effective than the MW treatment (Table 3). Our results were supported by the study of Andreani and Karboune [40] who compared enzymatic and microwave extraction approaches for the generation of oligosaccharides from American cranberry (*Vaccinium macrocapon*) pomace. In this investigation, higher carbohydrate yields were obtained when the cranberry pomace was treated by multi-enzymatic biocatalysts compared to the microwave-assisted extraction. Moreover, the enrichment of the lentil hulls in terms of soluble phenolic compounds by Pectinex^®^ Ultra Tropical could be explained by the breakdown of pectin into smaller polysaccharides and, to lower extent, cellulose and hemicellulose polymers, thereby allowing the phenolic compounds to be released from the cell wall matrix and increasing the radical scavenging activity. In previous studies, pectinase treatments (Pectinex Yield Mash and Pectinex AFP L-4, to a greater extent) of apple pomace clearly yielded a higher content of soluble phenolics and increased the radical scavenging activity of puree-enriched cloudy apple juices as compared to the untreated control sample [44], which supported our results.

The MW + EH-P treatment led to lower oligosaccharide and phenolic yields in the lentil hulls and decreased the antioxidant activity as compared to the enzyme treatment EH-P (Table 3, *p* ≤ 0.05). In contrast, the highest water-soluble fraction yield was observed with the sequential application of MW and EH-P (Table 3). Hence, as the oligosaccharide and phenolic yields could not explain the increasing difference in the solubilization yield as compared to the other treatments, we hypothesized that this increase was due to a higher solubilization of fiber components by the MW pretreatment and the subsequent hydrolysis by Pectinex Ultra Tropical to oligo-, di- and monosaccharides. Some studies have demonstrated that after microwave treatment, the surface area between enzyme and dietary fiber may increase, thus promoting penetration and the release of soluble substances [36]. The decreased yield of total phenolic compounds and antioxidant activity by the microwave pretreatments combined with enzymes (Viscozyme^®^) might be attributed to the high temperature and microwave radiation affecting the rate of phenolic extraction from the lentil hulls, the enzymatic oxidation and the polymerization processes. Heat from the microwaves causes the structural degradation of phenolic compounds. This results in steric obstruction of the enzyme binding sites to the substrate, which inhibits the degradation of cell-wall components [16].

### 3.5. Pectinex Treatment of Lentil Hull Increased the Content of Extractable Flavan-3-ols to a Greater Extent than Microwaves or Their Combined Treatment

The changes in the level of free phenolic compounds in the lentil hulls before and after the MW, EH-P and MW + EH-P treatments were also measured by HPLC-ESI-MS^2^. The phenolics were identified based on the MS^2^ data, UV spectra and retention times (Rt) compared with those of authentic standards, in a database and/or in the literature. A total of 18 compounds were identified or tentatively characterized in the different lentil hull samples. Detailed information of the identified compounds is summarized in Appendix A. The main phenolic compounds of the red lentil hulls were (+)-catechin *O*-hexoside (763 µg/g), galloylated dimer I (532 µg/g), procyanidin dimer II (482 µg/g), procyanidin trimer (453 µg/g), prodelphinidin dimer I (446 µg/g), prodelphinidin trimer (390 µg/g) and gallic acid (374 µg/g) (Table 4). (−)-epicatechin, quercetin and kaempferol glucosides and *trans-p*-coumaric acid derivatives were also detected in much lower amounts. The phenolic composition of the red lentil hulls was in accordance with previous studies that characterized the free phenolic profile of hulls from different lentil varieties (CDC Greenland, CDC 3494-6, CDC Invincible, CDC Maxim, Laird and Eston) [4,6]. In these subsets of commercial lentil varieties, the most abundant free phenolics were procyanidins, including monomeric, dimeric and trimeric flavan-3-ols.

Overall, the MW treatment sharply increased the levels of individual free phenolics such as (+)-catechin (236% vs. control sample) and, to a lesser extent, proanthocynidins: prodelphinidin dimer III, procyanidin dimer II and trimers of prodelphinidin and procyanidin (95%, 35%, 34% and 31% vs. control, respectively) (Table 4). Proanthocyanidins are an important group of bioactive molecules known for their promising benefits to human health [45]. Previous work has been reported on the microwave-assisted extraction of proanthocyanidin yields from seeds, peels, pomaces, leaves and barks of agro-industrial wastes [45,46], although there have been no studies performed for legume seed coats. Generally, in recent studies, higher recovery rates were reported for microwave-assisted extraction as compared with traditional extractions, although the extraction yields may vary depending on the processing conditions.

Compared to the control, EH-P enhanced the release of free phenolic compounds, mainly *trans*-*p*-coumaric acid derivative I, prodelphinidins, procyanidins and (+)-catechin (>79% vs. control) and (+)-catechin *O*-hexoside (26.6% vs. control), as shown in Table 4. As discussed in Section 3.3, the enhancement of polyphenols may result from structural changes in the food matrix due to the enzymatic hydrolysis of fibrous polymers in the lentil hull cell walls. Proanthocyanidins are also the main polyphenolic compounds that are retained in apple pomace because they readily bind to cell-wall polysaccharides through hydrogen bonding and/or hydrophobic interactions [47]. Similar to our results, Oszmianski et al. [48] demonstrated that enzymatic preparations containing pectinase (Pectinex Yield Mash, Pectinex Smash XXL and Pectinex XXL) increased the polymeric procyanidin contents in puree-enriched cloudy juices. This was likely due to the enzymatic degradation of both the cell wall and vacuolar membrane that enabled the optimal recovery of these compounds. In another study, the use of a commercial pectinase (Endozym^®^ Pectofruit PR) during the maceration of the winemaking process resulted in an increase in the extraction yield and the procyanidin B1 and B2 concentrations of grape juices [49].

The comparison of EH-P and MW indicated that Pectinex^®^ Ultra Tropical increased, to a higher extent, the solubilization of most of the phenolic compounds present in the lentil hulls including prodelphinidin dimer II, prodelphinidin trimer, galloylated dimer I, procyanidin dimer I, procyanidin trimer and (+)-catechin *O*-hexoside. Although an overall positive effect was observed in the free phenolic content of the lentil hulls upon enzymatic treatment, some phenolic acids (*trans-p*-coumaric acid derivative II) and flavonoids (quercetin rutinoside hexoside, kaempferol dirutinoside and kaempferol rutinoside hexoside) were significantly lower after the EH-P and EHP+MW treatments than the MW treatment (*p* ≤ 0.05). This observation could be due to new interactions between the phenolics (*trans-p*-coumaric acid derivative II, quercetin rutinoside hexoside, kaempferol dirutinoside and kaempferol rutinoside hexoside) and other enzymatic hydrolysis products, particularly oligosaccharides [50]. These results suggested that the hydrolysate matrices of the lentil hulls were enriched mainly in the free proanthocyanidins that prevailed as compared to phenolic acids and flavonoids.

Finally, the MW + EH-P treatment of the lentil hulls showed similar effects in the solubilization of phenolic compounds as compared to the EH-P treatment, although small differences were found between both treatments. MW + EH-P caused a lower extraction of prodelphinidin dimer II, galloylated dimer I and procyanidin dimer I (38, 20 and 23% decrease vs. EH-P, respectively), but slightly higher yields for procyanidin dimer II (15% increase vs. EH-P). Such differences could be explained by thermally induced polyphenol oxidation that may be responsible for the lower antioxidant activity obtained in the lentil hulls after the MW + EH-P treatment (Table 3). Studies have often indicated that some polyphenols are non-stable and sensitive to thermal treatment. Although microwave heating causes a high level of cell disintegration of plant tissues, flavan-3-ols are sensitive to thermal treatments. Fernandes et al. [51] demonstrated that 50% of flavan-3-ols in apple pomace were reduced with hot water extraction.

## 4. Conclusions

The comprehensive chemical characterization of industrial lentil hulls obtained from the production of football and split lentils reinforced the potential valorization of this underutilized by-product that is typically discarded as waste. Thus, the high content of fibers and total phenolic compounds described in this work warrants further functional studies dealing with the utility of lentil hulls as a human food ingredient. This study also highlighted the key role played by the selection of the commercial enzymatic preparation to streamline the potential food applications of lentil hulls. Thus, the screening of seven commercial enzymatic preparations revealed the suitability of Pectinex^®^ Ultra Tropical for releasing oligosaccharides and phenolic compounds from the lentil hull matrix and, consequently, increasing the antioxidant activity in the hydrolysates. The follow-up of the time course of the enzymatic reactions by the Pectinex^®^ Ultra Tropical indicated that, after 3 h, the maximum oligosaccharide and phenolic compounds yields as well as the highest antioxidant activity were achieved. The sequential microwave and enzymatic hydrolysis treatment, although it increased the solubilization yield of the lentil hulls, it decreased the content of oligosaccharides and proanthocyanidins and the antioxidant activity. Therefore, the enzymatic hydrolysis treatment alone was more suitable for producing a lentil hull hydrolysate enriched in potential prebiotics and antioxidant compounds. Future research should be focused on the optimization of the process by using different enzymatic hydrolysis parameters (enzyme concentration, enzyme–substrate ratio, pH, temperature, time and stirring speed) in order to establish the full biological potential of lentil hulls.

## Figures and Tables

**Figure 1 molecules-27-08458-f001:**
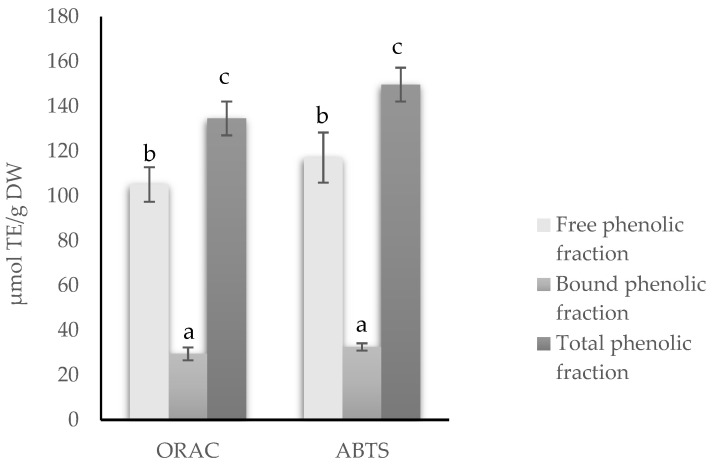
Antioxidant activity of free, bound and total phenolic fraction of red lentil hulls. Data are expressed as µmol Trolox equivalents (TE)/g dry weight. Error bars represent standard deviation (n = 6). Different letters denote statistical differences among free, bound and total phenolic fractions (*p* ≤ 0.05). Abbreviations: ABTS, 2,2′-azino-bis-3-ethylbenzothiazoline-6-sulfonic acid; ORAC, oxygen radical absorbance capacity; TE: Trolox equivalents; DW: dry weight.

**Figure 2 molecules-27-08458-f002:**
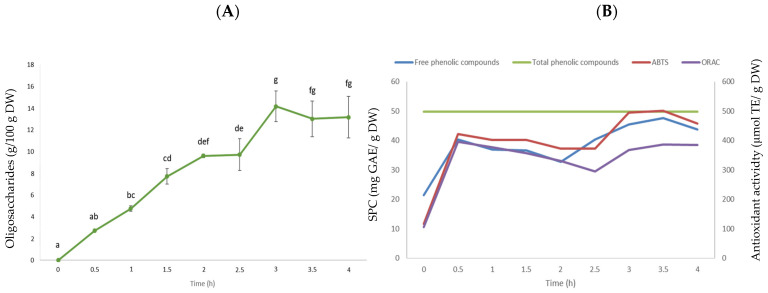
Time course of the extraction of oligosaccharides (**A**) and phenolic compounds and the increase in antioxidant activity as determined by ORAC and ABTS assays (**B**). Data are the mean ± standard deviation (n = 6). Different lowercase letters indicate statistical differences among each time (*p* ≤ 0.05 Bonferroni’s post hoc test). Green line in panel (**B**) represents the total phenolic content in untreated lentil hulls. Abbreviations: ABTS, 2,2′-azino-bis-3-ethylbenzothiazoline-6-sulfonic acid radical scavenging activity; ORAC, oxygen radical absorbance capacity; SPC, soluble phenolic compounds; GAE, gallic acid equivalents; TE, Trolox equivalents; DW, dry weight.

**Table 1 molecules-27-08458-t001:** Nutritional composition and total phenolic content of red lentil hulls from lentil processing industry (Prairie Pulse Inc., Vanscoy, SK, Canada).

Nutritional Parameters	Units	Mean ± SD ^1^
TDF	g/100 g	78.43 ± 2.13
IDF	g/100 g	69.32 ± 2.67
HMW-SDF	g/100 g	9.11 ± 0.55
Oligosaccharides	g/100 g	ND
Protein	g/100 g	9.12 ± 0.01
Starch	g/100 g	0.13 ± 0.01
Phytic acid	g/100 g	0.06 ± 0.00
Trypsin inhibitory activity	TIU/mg	18.26 ± 1.50
Total phenolic compounds	mg GAE/g	49.76 ± 4.74
Free phenolic compounds	mg GAE/g	21.44 ± 0.82
Bound phenolic compound	mg GAE/g	27.45 ± 2.62
Total Condensed tannins	mg CAE/g	15.83 ± 1.32

^1^ (n = 6). Abbreviations: CAE, catechin equivalents; GAE, gallic acid equivalents; HMW-SDF, high-molecular-weight soluble dietary fiber; IDF, insoluble dietary fiber; SD, standard deviation; TDF, total dietary fiber; ND: non-detected; TIU: trypsin inhibitory units.

**Table 2 molecules-27-08458-t002:** Oligosaccharides and soluble phenolic compound release and antioxidant activity in hydrolyzed lentil seed coats treated by seven commercial food-grade enzymes.

Enzymes	OS (g/100 g DW)	SPC (mg GAE/g DW)	SPC Ratio (%)	ORAC (µmol TE/g DW)	ABTS (µmol TE/g DW)
Control	ND	21.45 ± 2.62	-	105.11 ± 7.72	117.12 ± 11.17
Ultraflo^®^ XL	3.12 ± 0.15 ^a,^***	41.04 ± 1.25 ^ab,^*	71.39 ± 4.57 ^ab^	451.40 ± 10.85 ^b,^*	402.01 ± 37.07 ^b,^*
Ultraflo^®^ Max	1.89 ± 0.01 ^a^	38.29 ± 1.44 ^ab,^*	61.35 ± 5.24 ^ab^	399.93 ± 19.40 ^a,^*	390.82 ± 18.95 ^a,^*
Ultimase^®^ BWL 40	1.55 ± 0.17 ^a^	40.70 ± 0.37 ^ab,^*	70.14 ± 1.34 ^ab^	445.01 ± 9.79 ^b,^*	417.31 ± 43.18 ^b,^*
Viscozyme^®^ L	7.54 ± 0.03 ^b,^*	41.44 ± 0.62 ^bc,^*	72.83 ± 2.27 ^bc^	421.06 ± 14.56 ^b,^*	421.55 ± 34.28 ^b,^*
Celluclast^®^ 1.5L	3.57 ± 0.04 ^a,^**	39.81 ± 0.97 ^ab,^*	66.92 ± 3.55 ^ab^	408.08 ± 11.60 ^a,^*	384.80 ± 32.34 ^a,^*
Pectinex^®^ Ultra Tropical	14.17 ± 1.41 ^c,^*	45.51 ± 4.44 ^c,^*	98.59 ± 0.96 ^c^	442.04 ± 15.44 ^b,^*	424.06 ± 29.11 ^b,^*
Shearzyme^®^ Plus 2X	9.79 ± 0.66 ^b,^*	36.18 ± 0.21 ^a,^*	53.67 ± 0.75 ^a^	342.17 ± 34.09 ^a,^*	359.95 ± 17.17 ^a,^*

Data are the mean ± standard deviation (n = 6). Different letters within a column indicate statistical differences among the enzymes (*p* ≤ 0.05 Bonferroni’s post hoc test). An asterisk indicates significant differences between control and enzyme treatments (* *p* ≤ 0.000009; ** *p* ≤ 0.000707; *** *p* ≤ 0.002215 Dunnett’s post hoc test). Abbreviations: ABTS, 2,2′-azino-bis-3-ethylbenzothiazoline-6-sulfonic acid radical scavenging activity; GAE, gallic acid equivalents; ORAC, oxygen radical absorbance capacity; ND, not detected; OS, oligosaccharides; SPC, soluble phenolic compounds; TE, Trolox equivalents; DW: dry weight.

**Table 3 molecules-27-08458-t003:** Effect of MW, EH-P and MW + EH-P on oligosaccharides, soluble phenolics, soluble water fraction yields and antioxidant activity of lentil hulls.

Treatment	Oligosaccharides (g/100 g DW)	SPC (mg GAE/g DW)	Water-Soluble Fraction Yield (g/100 g DW)	ORAC (µmol TE/g DW)	ABTS (µmol TE/g DW)
Control	ND	21.45 ± 2.62 ^a^	9.58 ± 0.47 ^a^	105.11 ± 7.72 ^a^	117.12 ± 11.17 ^a^
MW	4.65 ± 0.23 ^a^	36.28 ± 0.60 ^b^	12.59 ± 0.93 ^b^	245.60 ± 5.65 ^b^	236.97 ± 19.27 ^b^
EH-P	14.17 ± 1.41 ^c^	45.51 ± 4.44 ^d^	24.92 ± 0.25 ^c^	442.04 ± 15.44 ^d^	424.06 ± 29.11 ^d^
MW + EH-P	12.12 ± 0.81 ^b^	41.45 ± 0.96 ^c^	34.27 ± 0.67 ^d^	403.98 ± 20.33 ^c^	376.98 ± 15.63 ^c^

Data are the mean ± standard deviation (n = 6). Different letters within a column indicate statistical differences (*p* ≤ 0.05 Bonferroni’s post hoc test). Abbreviations: GAE, gallic acid equivalents; ND, not detected; MW, microwave; EH-P, enzymatic hydrolysis by Pectinex^®^ Ultra Tropical; MW + EH-P, sequential microwave and enzymatic hydrolysis by Pectinex^®^ Ultra Tropical; ABTS, 2,2′-azino-bis-3-ethylbenzothiazoline-6-sulfonic acid scavenging activity; ORAC, oxygen radical absorbance capacity; SPC, soluble phenolic compounds; TE, Trolox equivalents; GAE, gallic acid equivalents; DW: dry weight.

**Table 4 molecules-27-08458-t004:** Effect of MW, EH-P and MW + EH-P on individual phenolic compounds (µg/g DW) of lentil hulls.

Compound	Control	MW	EH-P	MW + EH-P
*Phenolic acid*				
Gallic acid	373.93 ± 30.88 ^a^	464.02 ± 10.39 ^a^	403.44 ± 37.10 ^a^	507.18 ± 49.37 ^a^
*trans*-*p*-coumaric acid derivative I	7.45 ± 0.96 ^b^	7.17 ± 0.80 ^b^	20.98 ± 2.03 ^c^	t ^a^
*trans-p*-coumaric acid derivative II	17.53 ± 0.96 ^c^	15.06 ± 1.86 ^c^	7.84 ± 0.92 ^b^	t ^a^
*Prodelphinidin*				
Dimer prodelphinidin I	446.10 ± 14.92 ^a^	517.98 ± 40.45 ^a^	536.34 ± 15.70 ^a^	550.42 ± 26.96 ^a^
Dimer prodelphinidin II	85.48 ± 1.69 ^a^	156.01 ± 68.59 ^ab^	391.85 ± 9.99 ^c^	243.11 ± 6.65 ^b^
Dimer prodelphinidin III	51.04 ± 1.13 ^a^	99.98 ± 14.93 ^b^	115.01 ± 8.83 ^b^	116.78 ± 5.46 ^b^
Trimer prodelphinidin	389.8 ± 34.42 ^a^	525.96 ± 54.47 ^b^	660.25 ± 15.02 ^c^	550.87 ± 5.96 ^bc^
*Procyanidin*				
Galloylated dimer I	532.8 ± 49.12 ^a^	546.38 ± 10.02 ^a^	955.32 ± 32.57 ^c^	764.40 ± 43.70 ^b^
Galloylated dimer II	100.17 ± 20.93 ^a^	111.31 ± 30.51 ^a^	205.57 ± 11.87 ^b^	159.77 ± 7.20 ^ab^
Dimer procyanidin I	111.70 ± 21.67 ^a^	146.26 ± 15.28 ^a^	263.10 ± 2.12 ^c^	202.29 ± 16.23 ^b^
Dimer procyanidin II	481.45 ± 1.24 ^a^	648.39 ± 51.75 ^b^	760.26 ± 11.67 ^b^	872.67 ± 4.31 ^c^
Trimer procyanidin	452.84 ± 53.58 ^a^	594.95 ± 53.99 ^b^	974.60 ± 3.86 ^c^	912.56 ± 1.74 ^c^
*Flavonoid*				
(+)-catechin *O*-hexoside	762.7 ± 0.12 ^a^	799.62 ± 74.90 ^a^	965.40 ± 1.00 ^b^	957.21 ± 16.46 ^b^
(+)-catechin	164.78 ± 16.53 ^a^	553.83 ± 12.19 ^c^	426.83 ± 22.79 ^b^	416.01 ± 11.33 ^b^
(-)-epicatechin	158.28 ± 11.44 ^a^	179.61 ± 7.32 ^a^	211.64 ± 21.23 ^a^	165.27 ± 23.58 ^a^
Quercetin rutinoside hexoside	9.54 ± 0.35 ^c^	9.02 ± 0.41 ^c^	4.89 ± 0.05 ^a^	7.14 ± 0.63 ^b^
Kaempferol dirutinoside	59.62 ± 2.73 ^b^	59.16 ± 3.57 ^b^	20.74 ± 0.11 ^a^	25.88 ± 1.55 ^a^
Kaempferol rutinoside hexoside	58.54 ± 2.36 ^b^	54.55 ± 0.20 ^b^	29.41 ± 0.48 ^a^	32.27 ± 1.16 ^a^

Data are the mean ± standard deviation (n = 6). Different letters across columns in the same row indicate statistical difference (*p* ≤ 0.05 Bonferroni’s post hoc test). Abbreviation: MW, microwaves; EH-P, enzymatic hydrolysis by Pectinex^®^ Ultra Tropical; MW + EH-P, sequential microwaves and enzymatic hydrolysis by Pectinex^®^ Ultra Tropical; t, traces.

## Data Availability

Not applicable.

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
