# Peer review of "Selection of Enzymatic Treatments for Upcycling Lentil Hulls into Ingredients Rich in Oligosaccharides and Free Phenolics"

_molecules, 2022, doi:10.3390/molecules27238458_

Round 1

Reviewer 1 Report

The manuscript deals with the valorization of lentil hulls through enzymatic hydrolysis and MW extraction. The manuscript deals with a very interesting subject and should be published after minor revision.

Specific comments:

Title of the paper:

Microwaves and enzymatic hydrolysis: are they promising technologies for  valorization of lentil hulls as a functional ingredient?

The title is a bit inadequate, bearing in mind that MW extraction is not thoroughly optimized and accounts for a small part of overall results.

Methods

2.2. Enzymatic treatment

line 120- Authors should explain how they chose solid to solvent ratio since it plays a very important role in extraction efficiency.

line 124- There is no data about the final volume for the enzymatic treatment, so I can’t tell if the 2 ml aliquot would significantly affect the volume of the reaction.

3. Results

lines 311-312- Non-digestible oligosaccharides (LMW-SDF) were not detected in red lentil hulls (Table 1) in accordance with [2].

The reference isn’t adequately quoted.

line 320- I don’t understand how you got the ratio of 4.18 for IDF: SDF from the results presented in Table1.

lines 457-465- It would be easier to follow if the result with and without enzymatic hydrolysis were put together.

line -466- The authors should give an explanation for the increase in antioxidant activity since the results presented in Figure 1. state that bound phenolics have much lower antioxidant potential compared to free phenolics. What is then the explanation for this 3-4 fold increase if you release these bound phenolics with much lower antioxidant potential?

line 501-maximum of phenolic extraction is after 3.5 h. You can give a comment that after 3 h reasonable amount of phenolics was extracted and that oligosaccharides have their maxima, but it is not the maximum for phenolics according to your Figure.

line 634- You just explained why the yield of these compounds didn’t increase, but not why it is lower with EH-P and MW+EH-P.

Figures need to be corrected, maybe You could merge Figures 2 a and b since you follow the same kinetics. 

Reviewer 2 Report

The manuscript is interesting and reflects important work to add value to a by-product with potential health benefits.

Some suggestions are made

Line 124-125, 137-138: indicate at what temperature the enzyme was inactivated

Line 187: Specify at which wavelength it was measured.

Line 202: why chilled ethanol?

Line 313: in table 1 could also report total phenols

Describe in the methodology how the combined enzyme and microwave treatment was made, in which order?, What treatment did the sample have after one treatment to use the other?

Line 320-321: The sentence is not clear, how the radius between IDF and SDF is calculated?

Line 323-325: Explain why, since only 25% is accessible for fermentation by the gut microbiota

Line 351, 359, 360: Is it correct to describe as soluble phenols, according to the method used? Consider free and bound phenols as indicated in the table, or indicate in which it is soluble, since it will depend on the solvent used. Revise this expression through the manuscript

Line 369: add bound phenolics in title

Figure 1. Why was the statistic not performed with the total phenols?. Although it is a sum of the free and bound phenols, it is interesting to make the comparison alone and total, in addition, it can be correlated with the identification of phenols described later. Statistical comparison is recommended.

Figure 2. Improve the quality of the graphics, it is recommended to color the letters and numbers in black for a better visualization

Figure 2A. The concentration of total phenolic compounds was measured at each time?, this is reflected in the graph, however, the concentration of free phenols increases with time and the concentration of  total phenols was always the same over time, it is correct?. Check.

Line 570-578: The explanation is not very clear. It is necessary to explain from the methodology what was the sequence of the treatments, first MW and later Pectinex® Ultra Tropical or first Pectinex® Ultra Tropical and then MW. Because this paragraph can be confusing or contradictory. Microwave treatment is discussed first: “we hypothesize that this increase is due to the synergy of MW and enzymatic hydrolysis: as the carbohydrates were hydrolyzed to smaller soluble components (disaccharides and monosaccharides)...”. What enzyme are you referring to?  that of the treatment?. Then, it is understood that the MW treatment was carried out first and then the enzymatic one, verify if this was the case, if not, explain it to avoid confusion, since when applying the treatment with Pectinex® Ultra Tropical, in the end of the treatment the enzyme is inactivated or what enzyme are you referring?. Subsequently, it is discussed that the decreased yield of total phenolic compounds and antioxidant activity by microwave pretreatments combined with enzyme might be attributed to the high temperature and microwave radiation affecting the rate of phenolic extraction from the lentil hull, enzymatic oxidation, and polymerization processes…….cell-wall components [16]. Are you referring to the Pectinex® Ultra Tropical enzyme?. Improve this part of discussion for better understanding.

Line 650-651: Correct the citation form, change coworkers to et al

Avoid giving numerical results in the conclusion, these were already indicated in the results session
